# Short-Term Effects of Salt Restriction via Home Dishes Do Not Persist in the Long Term: A Randomized Control Study

**DOI:** 10.3390/nu12103034

**Published:** 2020-10-03

**Authors:** Sachiko Maruya, Ribeka Takachi, Maki Kanda, Misako Nakadate, Junko Ishihara

**Affiliations:** 1Department of Food Science and Nutrition, Faculty of Human Life and Environment, Nara Women’s University, Kitauoya-Nishimachi, Nara 630-8506, Japan; s-maruya@cc.nara-wu.ac.jp (S.M.); mkmg22856@gmail.com (M.K.); 2Department of Food and Life Science, Azabu University, 1-17-71 Fuchinobe, Chuo-ku, Sagamihara-city, Kanagawa 252-5201, Japan; nakadate@azabu-u.ac.jp (M.N.); j-ishihara@azabu-u.ac.jp (J.I.)

**Keywords:** sodium, food preference, randomized control trial

## Abstract

Salt intake reduction is crucial to prevent non-communicable diseases (NCDs) globally. This study aimed to investigate the short- and long-term effects of monitoring salt concentration in homemade dishes on reducing salt intake in a Japanese population. A double-blind randomized controlled trial using a 2 × 2 factorial design with two interventions was conducted in 195 participants; they were assigned to both interventions for a group monitoring salt concentration in soups (control: no monitoring) and a group using low-sodium seasoning (control: regular seasoning). We evaluated 24-hour urinary sodium excretions at baseline and after a three-month intervention for the changes as major outcomes, at six- and twelve-months after baseline as long-term follow-up surveys. Urinary sodium excretion decreased in both intervention and control groups after the intervention. However, differences in the change for both monitoring and low-sodium seasoning interventions were statistically non-significant (*p* = 0.29 and 0.52, respectively). Urinary sodium excretion returned to the baseline level after twelve-months for all groups. Monitoring of salt concentration is ineffective in reducing salt intake for short- and long-term among the people studied in this cohort.

## 1. Introduction

Excessive salt (i.e., total salt equivalent) intake is a risk factor for cardiovascular diseases, such as hypertension, coronary heart disease, and cerebral stroke, as well as gastric cancer, which are the leading causes of death globally [1,2]. Moreover, high salt intake causes early death and disability, as indicated by the disability-adjusted life year, as well as non-communicable diseases (NCDs) [3]. Thus, reducing salt intake is considered a priority globally [4].

Despite recommendations that salt intake should be reduced to less than 5 g/day by the World Health Organization (WHO), that is not practical in most populations, in many areas in the world like Europe, North America, and Asia [5,6]. Asian people, including Japanese, have the highest levels of salt intake in the world, which is an average of 12.7 g salt/day, measured by a 24-hour urinary sodium excretion, which reflects one’s intake of salt in a day [6]. The mean salt intake among Japanese is 10.0g, which is still twice the WHO recommendations [7]. Tailored salt reduction interventions have been reported to be successful in Japanese who are at high risk of stomach cancer and stroke due in part to excessive salt intake [8,9]. However, it is also necessary to establish successful interventions in healthy, low-risk populations in the effort to achieve primary prevention of NCDs.

The consumption of salt in traditional Asian diets is characterized by a relatively large contribution from discretional salty seasonings added during food preparation at home or at the table, and in particular, from soup or stew (16.4% of total sodium among Japanese adults and 21.8% among Korean adults) [10,11]. The use of a wide variety of salty seasonings, especially soy sauce and miso (i.e., fermented soybean paste) contribute greatly to salt intake [10]. Such discretional seasoning use is associated with individual’s taste preference [12]. Thus, given the primary sources of salt equivalents mentioned above, some approaches to increase individuals’ awareness on their saltiness preferences may be effective. Furthermore, the taste preference for miso soup as representative of homemade cooking was suggested to be a proxy index of daily sodium consumption [13]. This indicates that salt intake from home seasoning is a target for salt reduction. Monitoring salt concentrations by a salinity meter can be one of the methods for reducing salt intake via home seasoning as an approach to raising individuals’ awareness regarding their saltiness preferences for their dishes. We therefore hypothesized that when individuals monitor the salt concentration of their homemade dishes, they can become aware of their actual saltiness preference. This would lead to increased consciousness on salt intake, thereby facilitating overall reduction of salt intake.

In a previous short-term pilot study, we reported that monitoring salt concentrations of homemade dishes, which meant being aware of taste preferences, had a potentially stronger salt-reducing effect compared to the use of low-sodium seasonings (conventional method) [14]. Furthermore, other studies reported the effectiveness of low-sodium seasoning on reduced salt intake [15,16]. There are some studies that have reported the effects of self-monitoring on reducing salt intake [17,18]. However, their interventions measured sodium excretion in overnight (8-hours) urine, and they aimed to investigate the effects of individuals’ awareness of actual salt intake rather than their taste preferences. Moreover, collecting overnight urine and measuring salt concentration requires more effort compared to measuring only the salt concentration in dishes.

The aim of the present study was to further investigate the hypothesis mentioned above among people with an average level of salt intake in Japan. We examined whether the short-term effects of visual monitoring and use of low-sodium seasoning in reducing salt intake are reproducible as well as long-term effects after intervention. 

## 2. Materials and Methods

### 2.1. Participants

This randomized controlled trial was conducted by Sagami Women’s University, Nara Women’s University, and Azabu University, and study participants were recruited from healthy voluntary groups among adult residents, such as voluntary groups of Consumers’ Cooperative societies and clerical staffs of public health centers aged less than 75 years. We accepted the participants to join the study by their family units. The exclusion criteria were current use of hypertensive medicine and presence of soy allergy, and the criteria were formulated for all participants if they were in the same family. Consequently, we successfully recruited 200 participants. The study protocol was explained to the candidates through an orientation. A written informed consent was obtained from each participant. In total, 73 men and 127 women aged between 21 and 74 years old participated in the trial. Surveys were conducted between June 2015 and March 2018. The present study was registered in the University Hospital Medical Information Network Clinical Trials Registry (UMIN-CTR) under the Clinical Trial Registration System (UMIN000017773) and was approved by the institutional review boards of Sagami Women’s University (Approved on 12 May 2015; ethical code 14135), Nara Women’s University (Approved on 7 December 2015), and Azabu University (Approved on 17 April 2017).

### 2.2. Interventions

A randomized controlled trial was performed using a 2 × 2 factorial design with two interventions: home seasoning monitoring (monitoring) and using low-sodium seasoning (seasoning). We recruited 29–51 participants in each of five different time periods between June of 2015 and March of 2017. Participants who provided consent to be included in this study were allocated into the following four groups within each time period: (1) monitoring salt concentrations and using low-sodium seasoning, (2) monitoring salt concentrations and using regular seasoning, (3) not monitoring salt concentrations and using low-sodium seasoning, and (4) not monitoring salt concentrations and using regular seasoning. Family members of the same household were assigned to the same group. Therefore, with regard to monitoring intervention, the intervention group consisted of groups (1) and (2) and the control group consisted of groups (3) and (4). Similarly, for seasoning intervention, the intervention group consisted of groups (1) and (3) and the control group consisted of groups (2) and (4) (Figure 1). Three participants who missed urine collection ≥2 times during the 24-hour collection at baseline and two users of hypertensive medicine discovered during the baseline survey were excluded. Furthermore, participants who missed urine collection during follow-up surveys were excluded from the analysis of comparison to baseline (Figure 1). The random allocations were carried out independently by a research coordinator. The study was double-blinded so that neither the participants nor evaluators of outcomes knew the allocation of intervention during the intervention, evaluation, and analysis. In addition, regardless of group allocation, all participants were provided information on reducing salt intake at the start of the intervention through the distribution of a clear folder printed with tips on reducing salt intake and the amount of salt in some salty foods [19].

All participants were given instructions on the use of measuring instruments during the orientation. However, only participants in the monitoring intervention group were provided with the instrument prior to the intervention. They were not informed that monitoring by the instruments was a type of intervention or when the instruments would be provided to them. They were asked to measure and record salt concentrations in soup bowls or soup stock of stewed dishes (soups) at least once a week using a salinity meter (SK-5SII, Sato Keiryoki Mfg. Co., Ltd., Tokyo, Japan), if it was provided by the research coordinating office in the intervention period. Our rationale for requesting only once per week monitoring was that we did not want to unintentionally encourage participants to cook soup more than their usual soups preparation habits. Salt concentration measurements were recorded on a waterproof sheet and collected at the end of the three-month intervention. The instruments used did not necessarily need to be returned to the office after the intervention period. For the control group, a salinity meter was also provided after the follow-up surveys.

The seasoning intervention group received commercially available low-sodium seasoning (miso and soy sauce) from the research coordinating office; the control group received regular-sodium seasoning. The objective of the low-sodium seasoning intervention, a conventional method, was to prevent participants from becoming aware that salt monitoring was the primary intervention. Salt concentration of the low-sodium seasoning was approximately 7% (7 g salt/100 g) for both miso and soy sauce. The salt concentration in the regular-sodium seasoning was 11–12% (11–12 g salt/100 g) for miso and 14.5% (14.5 g salt/100 g) for soy sauce. There was no external packaging or salt concentration labeling in either of the groups. To blind the intervention, participants were not informed on the type of seasoning they received. Participants were asked to replace their usual miso or soy sauce with seasonings delivered monthly from the office when preparing dishes during the intervention period. We did not restrict their use of other seasonings. A questionnaire regarding the amount of residual seasoning and any reasons for discontinued use of the seasoning was administered to check participants’ compliance with interventions. After the three-month intervention period, delivered seasonings from the office need not be returned.

### 2.3. Outcome Evaluations

The primary outcomes were changes in 24-hour urinary sodium levels in a single excretion which reflects salt intake at baseline and three time points from baseline, which were as follows: at baseline, the end of a three-month intervention, six-months, and twelve-months as follow-up surveys. A 24-hour urine collecting device (U-container, Precise Urine Measurement Device, Sumitomo Bakelite Co., Ltd., Tokyo, Japan), which accurately collected 1/50 of the urine volume for 24 hours, was utilized [20]. We calculated 24-hour urinary excretion of sodium and intake of total salt equivalent (salt intake) by the following formula: 24-hour urinary excretion of sodium (mg/day) = obtained excretion (mL) × 50/1000 × urine sodium concentration (mEQ/L) × 23. Salt intake (g/day) = 24-hour urinary excretion of sodium (mg/day) × 2.54/1000. Information on alcohol intake, smoking history, current medication status, dietary habits including high-salt food or dishes using a questionnaire based on an existing questionnaire [21,22], and height and weight were self-reported by the participants.

### 2.4. Statistical Analysis

To calculate the sample size, we presumed a reduction in sodium by 1063 mg, based on the values demonstrated by a pilot study that examined the effect of monitoring sodium concentrations of homemade dishes. We computed that 198 subjects would be necessary to demonstrate a 1063 mg decrease in sodium intake, assuming an α error of 0.05, β error of 0.20, and dropout rate of 15% after randomization. 

We verified the normality of distribution with a histogram for each continuous variable. A chi-squared test and *t*-test were used to analyze baseline characteristics. Differences in the changes of urinary sodium excretion between the intervention and control groups were obtained using the following formula: change = post-intervention sodium − baseline sodium (mg/day) for all groups and difference of change = change in intervention group − change in control group (mg/day) for each intervention. Moreover, the statistical difference of changes following interventions between intervention and control groups were analyzed using *t*-tests and, as the least square mean, an analysis of covariance (ANCOVA) adjusted for sex (men or women); alcohol intake (none, occasionally, or every day); current medication status, except for hypertension drugs (no or yes); residential areas (Eastern region in Japan, Kanagawa Prefecture, and western Tokyo; or Western region in Japan, Nara Prefecture); received other interventions (no or yes); and interaction term of the interventions. Further adjustment was performed with the use of urinary sodium excretion level at baseline for the analysis of the low-sodium seasoning intervention. Additionally, we performed a stratified analysis by sex and whether the participants received seasoning intervention (for monitoring intervention analysis). IBM SPSS statistics version 21 (SPSS Inc., Chicago, IL, USA) and SAS version 9.4 for Windows (SAS Institute Inc., Cary, NC, USA) were used for analyses. The significance level was set at *p* < 0.05.

## 3. Results

In this study, the mean value (standard deviation) of urinary sodium excretion for all participants (*n* = 195) at baseline was 3907 (1515) mg/day. Baseline characteristics of the participants by each intervention are listed in Table 1. Age, smoking history, weight, and dietary habits showed no significant differences between the groups in either of the interventions. Non-significant differences were observed for sex, alcohol drinking, current medication status, and residential areas between the intervention and control groups in both monitoring and seasoning interventions. In addition, significant differences were not observed for the salty dietary habits between the intervention and control groups at the end of the intervention period surveys (data not shown). There was a statistically significant difference in the urinary sodium excretion for the seasoning intervention (control group, 4166 (1723) mg/day; intervention group, 3675 (1265) mg/day; *p* = 0.03).

Changes in urinary sodium excretion following monitoring and seasoning interventions and the differences between interventions and controls are illustrated in Table 2. With regard to the monitoring intervention, at the end of the three-month intervention, the reduction of urinary sodium excretion in terms of crude mean value in the intervention group was less than that in the control group, although not statistically significantly (−255 and −475 mg/day for intervention and control groups, respectively; *p* = 0.37). The result did not substantially change in the analysis adjusted for sex, alcohol drinking, medication status, residential areas, other interventions, and interaction term (−132 and −384 mg/day for intervention and control groups, respectively; *p* = 0.29). Changes from baseline decreased in six- and twelve-months follow-up surveys in both the intervention and control groups, with corresponding values of −324 and −33 mg/day in the intervention group, respectively, and −201 and −138 mg/day in the control group, respectively. Although the mean values of urinary sodium excretion were decreased after the intervention period in both groups, they returned to the baseline level in the long-term follow-up period (Figure 2a).

With regard to seasoning intervention, at the end of the three-month intervention, the reduction of urinary sodium excretion in terms of crude mean value in the intervention group was smaller than that in the control group without statistical significance (−287 and −463 mg/day for intervention and control groups, respectively; *p* = 0.47). The results were adjusted for sex, alcohol drinking, medication status, residential areas, other interventions, urinary sodium excretion level at baseline, and interaction term. The reduction of urinary sodium excretion in the intervention group was greater compared to that in control group although it was not statistically significant (−297 and −173 mg/day for intervention and control groups, respectively; *p* = 0.52). Changes from baseline decreased in the six- and twelve-month follow-up surveys in both intervention and control groups, with corresponding values of −139 and 6 mg/day in the intervention group, respectively, and −398 and −184 mg/day in the control group, respectively. Although the mean values of urinary sodium excretion decreased after the intervention period in both groups, they returned to baseline level in the long-term follow-up period (Figure 2b).

Neither intervention changed salt intake for long-term among consumers of general salt intake levels.

In the results of ANCOVA, there were no interactions between the monitoring and seasoning interventions at the end of intervention (*p* = 0.61 for monitoring interventions and *p* = 0.68 for low-sodium seasoning interventions, respectively; data shown in Appendix A
Appendix A).

The results for the stratified analysis by sex are illustrated in Table 3. Regarding the results in women, at the end of the three-month intervention, the reduction of urinary sodium excretion in terms of crude mean value in the intervention group was greater than that in the control group without statistical significance (−270 and −188 mg/day for monitoring intervention and the control groups, respectively; *p* = 0.76; −262 and −188 mg/day for seasoning intervention and the control groups, respectively; *p* = 0.78). The differences were also not significant when adjusted for potential confounders. In contrast, the reductions of urinary sodium excretion in the intervention groups were not greater than those of control groups for either interventions in both crude and least square means adjusted for possible confounders among men.

The results of the stratified analysis regarding monitoring intervention by whether received or not seasoning intervention are shown in Table 4. At the end of the three-month intervention, the decrease in the crude mean of urinary sodium excretion in the intervention group was smaller than that in the control group; however, this result was not statistically significant either in subjects who received seasoning intervention (−438 and −485 mg/day for the intervention and the control groups, respectively; *p* = 0.90) or in those that did not (−104 and −466 mg/day for the intervention and the control groups, respectively; *p* = 0.27). The differences were also not significant when adjusted for potential confounders. In the other stratified analyses (age, smoking history, alcohol intake, medication status, residential areas, weight, and urinary sodium excretion at baseline), there was no significant difference observed between the intervention and control groups for both monitoring and seasoning intervention (data not shown).

## 4. Discussion

Based on the results, there was no significant difference between the urinary sodium reductions of the intervention and control groups in either three-month monitoring or seasoning intervention. The reduction in the intervention group was greater than that in the control group for both interventions in women only. However, the differences were not significant. In the result of six- and twelve-month (after baseline) follow-up surveys following the intervention, urinary sodium excretion levels returned to baseline for all groups.

In this study, the overall urinary sodium excretion at baseline (9.9 g salt/24-hour urinary excretion) was lower than in previous studies. In our previous pilot study, 50 participants (15 men and 35 women) with high salt intake (11.5 g salt/24-hour urinary excretion) were subjected to home seasoning monitoring and low-sodium seasoning use interventions for three months [14], as in the present study. As a result, the reduction in urinary sodium excretion was greater in the intervention group than that in the control group, without statistical significance (the differences between the intervention and control groups were −1063 and −559 mg/day for home seasoning monitoring and low-sodium seasoning intervention, respectively). Another previous study that examined the effect of low-sodium seasoning among 17 subjects (one man and 16 women) with a relatively high salt intake (11.1 g salt/24-hour urinary excretion) reported that low-sodium seasoning intervention for a week resulted in significantly greater reduction of salt in urinary sodium excretion compared to control group (the difference between the intervention and control group was −1142 mg/day) [15]. Although they were small-scale studies, inconsistently with our results, the effect of monitoring or seasoning interventions was demonstrated. That suggests that monitoring or low-sodium seasoning intervention may be effective among subjects with relatively high salt-intake in Japan. In this study, 24-hour urinary sodium excretion decreased after intervention in control groups for both monitoring and seasoning interventions as well as in intervention groups. The result was similar in the analysis stratified by low-sodium seasoning intervention arms, in addition to their no-interaction effects. This matched our pilot study. However, 24-hour urinary sodium excretion did not decrease for the control group in a previous low-sodium seasoning sole intervention study [15]. Difference of decrement might be potentially under-estimated by participants of another intervention arms in the control group because of 2 × 2 factorial design. The salt intake level of the present study participants was close to the mean Japanese salt intake [7] and lower than in previous studies. The survey was conducted in five regions from three prefectures; therefore, the population included in this study was a better reflection of the general Japanese population.

In addition to the aspect of salt intake level, the percentage of women included in this study was smaller compared to that of previous studies. In the stratified analyses of both interventions, urinary sodium excretion decreased at a higher rate in the intervention group among women without statistical significance, which was consistent with previous studies. As a result that, in general, women are expected to manage the cooking in their household, they are more likely to have observed the salt concentration measurements of soups compared to men. Therefore, the effects of the monitoring intervention might be limited to women. Thus, it is necessary to consider adaptation of these interventions for gender characteristics.

We acquired information about the dietary habit including frequency and amount of salty food intake, and there was no difference between the control and intervention groups for both monitoring and seasoning interventions, at baseline and the end of the intervention.

In this study, we asked participants to monitor and record the salt concentration at least once a week, and more than 80% of participants in the monitoring intervention group measured and recorded data completely (12 times). Regarding seasoning intervention, 92% of participants in intervention group used provided seasonings throughout the three-month intervention period. In contrast, control group participants achieved 100%. We conducted intention-to-treat analysis for all allocated members as intervention groups, even though some participants did not measure their salt concentrations or stopped using the provided seasonings. Thus, the potential underestimation of the true magnitude of the effect of monitoring intervention cannot be excluded. Regarding seasoning intervention, the difference in compliance between the intervention and control groups could explain the present results. With respect to the concern of seasonal changing in dietary intakes, in this study, baseline survey and random allocation were conducted in five different periods. Therefore, although dietary and salt intakes are likely influenced by seasonal changing [21,23], their effects might be negligible.

The strength of the present study was the effects of home seasoning monitoring and low-sodium seasoning interventions, which could be followed up in six months and twelve months after baseline. In the present study, interventions were performed for three months; however, no long-term continuous impact was observed at either six- or twelve-month follow-up surveys. This result demonstrates the need for some consistent approaches.

There are some limitations in this study. First, urinary sodium excretion was only assessed once. The accuracy for estimations of salt intake with 24-hour urine collection is higher when measurements are performed multiple times as opposed to once [24]. However, the sample size of the present study was calculated based on the collection of 24-hour urine sample once daily [14], and the error had been considered in advance. Second, participants in this study were recruited from administrative agency clerical staffs as public health center and voluntary groups of Consumers’ Cooperative societies. Therefore, they might be health-conscious and might have relatively higher motivation for reducing their salt intake, although their salt intake amount was almost the same as the mean of Japanese people [7]. Third, we performed the present study using a 2 × 2 factorial design. Control groups for each intervention included those who received other interventions. Although we adjusted for interaction in interventions and whether received other intervention or not in ANCOVA, residual confounding is possible. Furthermore, unmeasured variables such as socioeconomic status could remain as confounders. However, the characteristics of participants among groups could be considered to equal because of the random allocation. Fourth, in this study, we accepted that subjects participated in family unit. As a result of this, there is a possibility that dietary behaviors of a person who managed the cooking in their family influenced their family members. If so, observed reduction in both groups of interventions and controls might be overestimated, although the difference of change between groups might not be influenced for the random allocation by family unit. Further studies are warranted on the difference between gender or manage and do not manage daily cooking.

Based on the findings, in previous pilot and present studies, interventions involving home seasoning monitoring and the use of low-sodium seasoning may have less impact on salt reduction than expected and suggest the possibility that they are not effective in populations with salt intake close to the mean salt intake of the Japanese population. However, the results of present study cannot exclude the possibility of effect on high-salt-intake population and/or women.

In conclusion, neither intervention (monitoring home seasoning nor low-sodium seasoning) showed short-term greater reduction on salt intake among people with average level of salt consumption in Japan. Moreover, urinary sodium excretion returned to the baseline level in follow-up surveys after a short-term intervention period.

## Figures and Tables

**Figure 1 nutrients-12-03034-f001:**
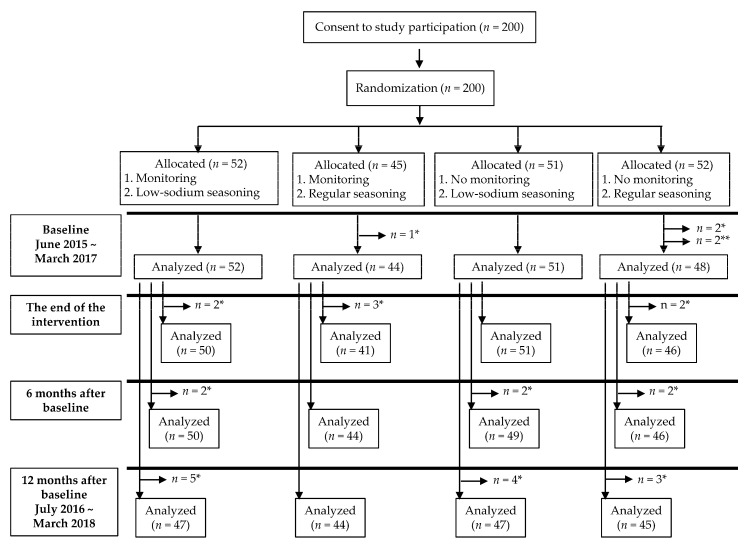
Study scheme. * Excluded for incomplete urine collection. ** Excluded for use of hypertensive medicine.

**Figure 2 nutrients-12-03034-f002:**
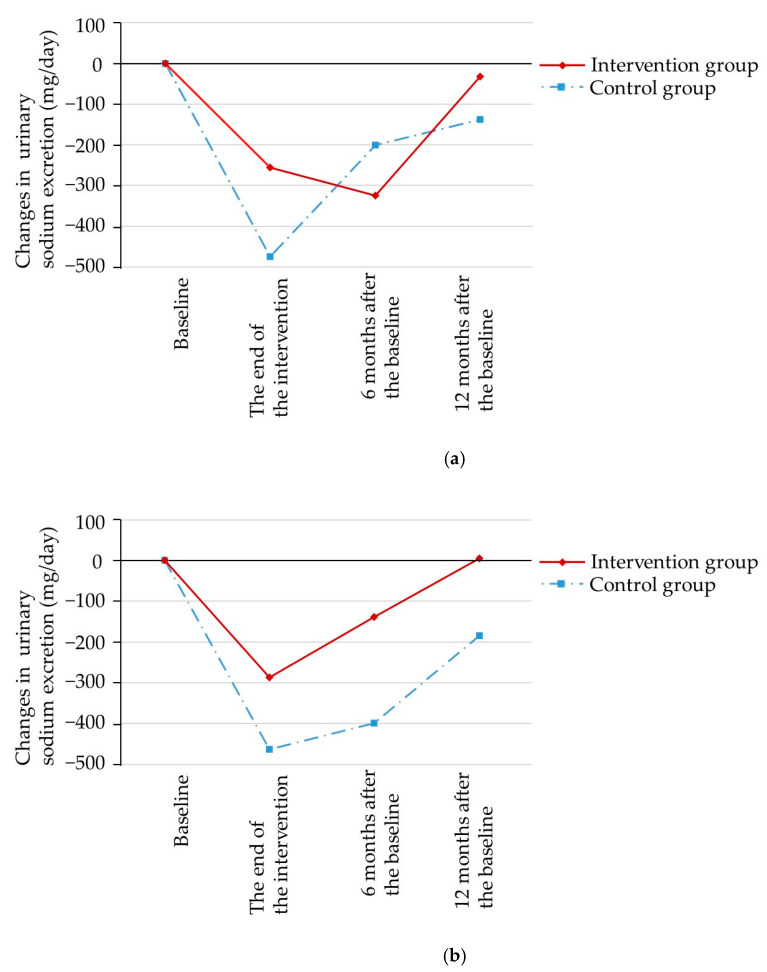
Changes in urinary sodium excretion for each intervention (mean based on crude value, taken from Table 2). (**a**): Monitoring intervention, (**b**): Low-sodium seasoning intervention.

**Table 1 nutrients-12-03034-t001:** Baseline characteristics of participants.

	Monitoring Intervention			Seasoning Intervention		
	Control Group (*n* = 99)	Intervention Group (*n* = 96)	*p* ^1)^	Control Group (*n* = 92)	Intervention Group (*n* = 103)	*p* ^1)^
Sex (% females)	62.6	67.7	0.46	62.0	68.0	0.38
Age ^2)^	47.2 (11.3)	47.8 (10.3)	0.68	47.6 (10.7)	47.4 (10.9)	0.92
**Smoking history (%)**			0.79			0.77
None	77.8	75.0		78.3	74.8	
Prior smoking history	15.2	18.8		16.3	17.5	
Current smoker	7.0	6.3		5.4	7.8	
**Alcohol intake (%)**			0.72			0.47
None	23.2	28.1		25.0	26.2	
Occasionally	45.5	43.8		48.9	40.8	
Every day	31.3	28.1		26.1	33.0	
Current medication status (% currently) ^3)^	23.2	14.6	0.12	22.8	15.5	0.20
**Residential area (%)**			0.19			0.56
Eastern region in Japan	57.1	42.9		44.2	55.9	
Western region in Japan	46.6	53.4		49.2	50.9	
Weight (kg) ^2)^	59.0 (12.7)	58.9 (11.0)	0.95	59.4 (12.2)	58.6 (11.6)	0.64
Urinary sodium excretion (mg/day) ^2)^	3908 (1581)	3,905 (1452)	0.99	4166 (1723)	3675 (1265)	0.03
**Dietary habits**						
miso soup (% 5 bowls or more /week)	62.6	60.4	0.75	60.8	62.1	0.86
taste preference (% prefer to moderately or strong taste)	9.1	4.1	0.17	7.6	5.8	0.62
noodle soup (% take one third or more)	75.8	63.4	0.06	71.7	68.0	0.57
pickles (% 3 times or more /week)	32.3	30.2	0.75	38.0	25.2	0.05
soy sauce using at table (% using)	57.6	58.3	0.91	60.9	55.3	0.44
salt using at table (% using)	32.3	32.3	1.00	32.6	32.0	0.93

^1)^*t*-tests for age, weight, and urinary sodium excretion. Chi-squared test for sex, smoking history, alcohol intake, current medication status, residential area, and dietary habits. ^2)^ Mean (standard deviation, SD); ^3)^ excluding hypertension drugs.

**Table 2 nutrients-12-03034-t002:** Changes in urinary sodium excretion, and differences between the intervention and control groups (mg/day).

	Intervention Group					Control Group					Difference of Change ^4)^	*p* ^5)^	Difference of Change ^6)^	*p* ^7)^
	*n*	Baseline ^1)^	Post Intervention ^1)^	Change ^2)^	Change ^3)^	*n*	Baseline ^1)^	Post Intervention ^1)^	Change ^2)^	Change ^3)^				
Monitoring intervention														
At the end of the intervention	91	4004	3750	−255	−132 ^8)^	97	3887	3412	−475	−384 ^8)^	220	0.37	253 ^8)^	0.29
		(1422)	(1449)	(−571, 62)	(−561, 297)		(1590)	(1313)	(−837, −112)	(−763, −5)	(−261, 701)		(−235, 741)	
6 months after the baseline	94	3950	3626	−324	−242 ^8)^	95	3860	3659	−201	−139 ^8)^	−123	0.64	−103 ^8)^	0.72
		(1431)	(1428)	(−657, 9)	(−690, 206)		(1570)	(1542)	(−584, 182)	(−546, 267)	(−627, 382)		(−619, 413)	
12 months after the baseline	91	3931	3898	-33	10 ^8)^	92	3874	3736	−138	−76 ^8)^	105	0.65	86 ^8)^	0.71
		(1448)	(147)	(−314, 248)	(−394, 415)		(1604)	(1487)	(−499, 222)	(−443, 291)	(−349, 560)		(−384, 557)	
Seasoning intervention														
At the end of the intervention	101	3714	3427	−287	−297 ^9)^	87	4210	3747	−463	−173 ^9)^	176	0.47	−123 ^9)^	0.52
		(1244)	(1384)	(−607, 34)	(−617, 23)		(1735)	(1379)	(−830, −95)	(−492, 146)	(−306, 656)		(−512, 264)	
6 months after the baseline	99	3696	3557	−139	−194 ^9)^	90	4134	3736	−398	−166 ^9)^	259	0.31	−28 ^9)^	0.89
		(1241)	(1528)	(−463, 185)	(−545, 157)		(1717)	(1432)	(−793, −3)	(−510, 178)	(−246,763)		(−450, 393)	
12 months after the baseline	94	3660	3666	6	−54 ^9)^	89	4159	3975	−184	9 ^9)^	190	0.41	−64 ^9)^	0.76
		(1247)	(1352)	(−323, 336)	(−394, 285)		(1742)	(1746)	(−499, 131)	(−319, 338)	(−264, 644)		(−473, 345)	

^1)^ Mean (SD); ^2)^ mean (95% confidence interval, CI), change = post-intervention urinary sodium excretion − baseline urinary sodium excretion (mg/day); ^3)^ least square, LS means (95% CI), change = post-intervention urinary sodium excretion − baseline urinary sodium excretion (mg/day); ^4)^ mean (95% CI), difference of change = change in intervention group − change in control group; ^5)^
*t*-tests; ^6)^ LS means (95% CI), difference of change = change in intervention group − change in control group; ^7)^ analysis of covariance; ^8)^ adjusted for sex, alcohol drinking habits, current medication status, residential areas, other intervention, and interaction term of the interventions; ^9)^ adjusted for sex, alcohol drinking habits, current medication status, residential areas, other intervention, baseline urinary sodium excretion, and interaction term of the interventions.

**Table 3 nutrients-12-03034-t003:** Changes in urinary sodium excretion and differences between the intervention and control groups by stratified analysis (mg/day).

	Intervention Group					Control Group					Difference of Change ^4)^	*p* ^5)^	Difference of Change ^6)^	*p* ^7)^
	*n*	Baseline ^1)^	Post Intervention ^1)^	Change ^2)^	Change ^3)^	*n*	Baseline ^1)^	Post Intervention ^1)^	Change ^2)^	Change ^3)^				
Men	30	4646	4422	−224	271 ^8)^	36	4533	3573	−960	−716 ^8)^	736	0.12	987 ^8)^	0.07
		(1661)	(1465)	(−828, 381)	(−747, 1289)		(2020)	(1327)	(−1702, −219)	(−1479, 47)	(−228, 1701)		(−90, 2064)	
Women	61	3688	3419	−270	−51 ^8)^	61	3505	3317	−188	3 ^8)^	−82	0.76	−54 ^8)^	0.91
		(1179)	(1332)	(−650, 110)	(−523, 421)		(1124)	(1305)	(−563, 187)	(−428, 433)	(−610, 447)		(−589, 482)	
Men	33	4187	3849	−338	−311 ^9)^	33	4982	4069	−913	−437 ^9)^	575	0.24	126 ^9)^	0.69
		(1293)	(1304)	−940, 265)	(−954, 332)		(2230)	(1587)	(−1694, −132)	(−1012, 139)	(−392, 1543)		(−580, 832)	
Women	68	3484	3222	−262	−178 ^9)^	54	3738	3551	−188	22 ^9)^	−74	0.78	−199 ^9)^	0.40
		(1161)	(1384)	(−649, 125)	(−560, 205)		(1133)	(1209)	(−543, 167)	(−389, 432)	(−606, 458)		(−676, 278)	

^1)^ Mean (SD); ^2)^ mean (95% CI), change = post-intervention urinary sodium excretion − baseline urinary sodium excretion (mg/day); ^3)^ LS means (95% CI), change = post-intervention urinary sodium excretion − baseline urinary sodium excretion (mg/day); ^4)^ mean (95% CI), difference of change = change in intervention group − change in control group; ^5)^
*t*-test; ^6)^ LS means (95% CI), difference of change = change in intervention group − change in control group; ^7)^ analysis of covariance; ^8)^ adjusted for alcohol intake, current medication status, residential areas, other intervention, and interaction term of the interventions; ^9)^ adjusted for alcohol intake, current medication status, residential areas, other intervention, baseline urinary sodium excretion, and interaction term of the interventions.

**Table 4 nutrients-12-03034-t004:** Changes in urinary sodium excretion, and differences between the intervention and control groups for monitoring intervention by stratified whether received seasoning intervention (mg/day).

	Intervention Group					Control Group					Difference of Change ^4)^	*p* ^5)^	Difference of Change ^6)^	*p* ^7)^
	*n*	Baseline ^1)^	Post Intervention ^1)^	Change ^2)^	Change ^3)^	*n*	Baseline ^1)^	Post Intervention ^1)^	Change ^2)^	Change ^3)^				
At the end of the intervention	41	4364	3925	−438	−303	46	4072	3588	−485	−453	46	0.90	150	0.69
		(1614)	(1465)	(−12, 864)	(−923, 318)		(1843)	(1293)	(−1083, 114)	(−1009, 104)	(−679, 772)		(−604, 905)	
6 months after the baseline	44	4227	3645	−583	−431	46	4045	3824	−221	−168	−361	0.36	−263	0.51
		(1639)	(1421)	(−61, 1104)	(−1073, 211)		(1802)	(1454)	(−827, 385)	(−767, 431)	(−1150, 427)		(−1053, 527)	
12 months after the baseline	44	4227	4072	−155	−92	45	4092	3880	−212	−87	57	0.86	−5	0.99
		(1639)	(1986)	(−244, 554)	(−615, 431)		(1854)	(1491)	(−690, 577)	(−574, 400)	(−576, 689)		(−650, 640)	
At the end of the intervention	50	3709	3605	−104	54	51	3718.32	3253	−466	−332	362	0.27	386	0.25
		(1177)	(1435)	(−364, 572)	(−576, 684)		(1317)	(1323)	(−915, −16)	(−879, 215)	(−279, 1003)		(−282, 1055)	
6 months after the baseline	50	3705	3608	−96	−144	49	3687	3505	−183	-85	86	0.79	59	0.86
		(1182)	(1448)	(−338, 530)	(−701, 421)		(1311)	(1626)	(−682, 312)	(−725, 558)	(−570, 780)		(−627, 747)	
12 months after the baseline	47	3735	3653	81	82	47	3667	3598	−68	−82	145	0.66	164	0.65
		(1212)	(1196)	(−488, 326)	(−568, 732)		(1309)	(1487)	(−603, 466)	(−653, 489)	(−514, 813)		(−542, 869)	

^1)^ Mean (SD); ^2)^ mean (95% CI), change = post-intervention urinary sodium excretion − baseline urinary sodium excretion (mg/day); ^3)^ LS means (95% CI), change = post-intervention urinary sodium excretion − baseline urinary sodium excretion (mg/day); ^4)^ mean (95% CI), difference of change = change in intervention group − change in control group; ^5)^
*t*-tests; ^6)^ LS means adjusted for sex, alcohol intake, current medication status, and residential areas (95% CI), difference of change = change in intervention group – change in control group; ^7)^ analysis of covariance.

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
