# Peer review of "Short-Term Effects of Salt Restriction via Home Dishes Do Not Persist in the Long Term: A Randomized Control Study"

_nutrients, 2020, doi:10.3390/nu12103034_

Round 1

Reviewer 1 Report

None

Author Response

Thank you for the time and energy you spent to review this work.

Reviewer 2 Report

Line 41: The word “and” should be omitted; it should be “… among Japanese is 10.8g [8], which is still…”

Lines 42-45:  The sentences that begin, “Some successful….” are awkward.   Do you mean to state the following?  “Tailored salt reduction interventions have been shown to be successful in Japanese who are at high risk of stomach cancer and stroke, due in part to excessive salt intake. However, it is also necessary to establish successful interventions in healthy, low risk populations in the effort to address prevention of non-communicable diseases.”

I’m not sure if that is the message you are attempting to convey, but the sentences need to be edited to be more concise and clear.

Lines 46-50:  Lengthy sentence that would read better if broken down into 2 sentences.

Line 51: “…individuals’ awareness of the salty taste based on salt concentration of the soup…”  would improve the sentence structure.  OR, “…individuals’ awareness of the salty taste of their foods…”

Line 55:  could be rewritten to improve sentence structure, “… one of the methods that could aid in reducing salt intake via home seasoning as an approach to raise individuals’ awareness of salty taste of foods.”

Line 57:  recommend, “…the salty taste of the soup.”

Line 58-59:  Has this been shown or is this part of your hypothesis?

Lines 115-116: This might be more clear and concise, “They were not informed that the monitoring instrument was an intervention.”

I’m not sure what you mean by, “...and when the instruments was reached.”

Lines 119-120: “Our rationale for requesting only once per week monitoring was that we did not want to unintentionally encourage participants to cook soup more than their usual soup preparation habits.”

Lines 127:  omit “as placebo”

Line 298:  change “measured” to “measuring”

Line 299:  add “salt” before “concentration”

Lines 299-300:  awkward – do you mean, “… can not be excluded.”

Lines 300-303:  The sentences that begin, “Concerning seasonal changing…”  is awkward.  I’m not sure what you are trying to state.

Lines 317-319:  The sentence beginning “Third, although it is possible…” is awkward. 

Reviewer 3 Report

The manuscript by Maruya et al. seeks to compare two interventions to reduce sodium intake over 3 months in Japanese adults. Additionally, subjects were followed for 9 months after the intervention. It is relatively large intervention study that was well controlled. One of the largest issues is the need for editing for the English language. There are areas where clarity is missing on the protocol and this likely stems from a language barrier. Therefore, it is highly recommended that a native English speaker carefully reviews this manuscript. The authors need to better explain why the changes in sodium intake seem to be greater in the control compared to intervention group as shown in figure 2. One would expect that group 4 who received no monitoring and used the regular seasoning would not see any changes in their urinary sodium excretion yet they did. The authors do not seem to address the limitations of their design and that their control groups did not differ from the intervention. This should be included in the discussion. Furthermore, what compliance measures were put into place to determine if subjects followed their respective interventions.

Abstract

Line 15 Consider revising “adjustment”

Line 62 This previous pilot study is by your group. Not clear why that is not stated in the paper

There is no rationale provided as to why salt reduction is necessary. Consider adding one line to support importance of area

Line 21-22 I think greater clarity is needed for the timepoints of this study. It was 3 months of an intervention followed by follow-up at 6 months and 12 months post-baseline

Intro

Line 34. Myocardial infarction is a result of heart disease, not necessarily a disease itself

Line 36 It is adjusted, not adjuster

Line 40-41 The use of two different stats on the salt intake of Japanese is confusing. Generally excretion is a reflection of intake. Consider explaining this better or give one stat.

Line 41. This sentence needs editing “… intake among Japanese is 10.0g and, which is still greater …”

Line 42-44 sentence needs extensive editing. Lacks clarity

Line 44-45. sentence needs extensive editing. Lacks clarity. Not clear what authors mean here

Include hypothesis at end of introduction

Methods

Was the age range 18-74?

The timepoints are not clear. It would be easier if all timepoints were identified from baseline. Hence, 3 months after intervention is really 6 month timepoint

Line 77-79 Some of this power analysis information could be moved to the stats section

Line 116 what does “when the instruments was reached” mean?

Line 123 it’s meter, not mater

Line 129, 131. Helpful to provide salt in g in addition to %

Line 140 urinary sodium excretion is a surrogate for sodium intake, not salt

Line 144-146 Were specific questionnaires used to gather this information?

Results

Overall, tables need to have data all on same line as opposed to below it. Word should not be split and rolling over to the next line. Some column titles are bolded while others are not.

Figures are blurry. The quality of the image needs to be improved.

Line 166 it is “are illustrated”, not “were”

Urinary sodium data at baseline is listed in table 1. Do not list again in text

Line 179 it is “are illustrated”, not “were”

Line 181 should define what you mean by crude mean value of sodium excretion in your methods or at least in how you planned to report the data

Line 186 get rid of “have”

Table 2. Why are some column titles bolded and others are not? Row titles do not line up with data. Hard to know what we are looking at

Figure 2 is a bit faint and blurry. The y axis title should not extend below the x axis line

Table 3. Not sure if this table is necessary as readers are just looking at the statistical output of your analysis. Given that very little is significant, it seems unnecessary to include and could be provided as a supplemental file

Table 4. First column is pushing “n” from women into line below. Some column headings are bolded but others are not

Discussion

Discussion

Not addressing similar changes in control groups to intervention

Line320 The conclusion should be clear that  neither intervention significantly reduced salt intake. The use of the “may have less impact on salt reduction than expected”. It is really than hypothesized. And then the conclusion that it may be more effective in women is related to a power issue. The study predominantly contained women. The study was not designed to assess sex differences.

Reviewer 4 Report

The manuscript entitled „Long-term effect of using low-sodium seasonings or monitoring salt concentration of dishes prepared at home on the reduction of sodium intake after a three-month intervention among healthy adult men and women: A randomized controlled trial in Japan” presents interesting issue but it should be corrected.

Major:

There are the number of major limitations of the study:

  1. Authors included patients within the same household to the study, while it may be supposed that they have a similar approach for the used seasonings, so within 1 household only participant should be included
  2. It seems that Authors controlled only some of the salt-containing seasonings (but it is not clearly stated by Authors), so it is the important bias
  3. Authors (in theory) randomly allocated participants to groups, but in fact there was different number of participants in the studied group – it is hard to understand how did it happen.
  4. Based on the described methodology it seems that Authors controlled sodium intake only in some dishes – if so, it is the other important source of bias.

General:

The manuscript is shabbily prepared with some comments in tracking changes option. Moreover, the manuscript should be prepared according to the instructions for authors.

Whole manuscript must be carefully corrected to be perfectly understandable for readers, as for the time being it is even not clarified if participants had the kitchen salt  with reduced sodium amount, soy sauce and miso sauce with reduced sodium amount, other sauces with reduced sodium amount, or all of them.

Authors should correct in the presented manuscript the applied vocabulary and their approach for the description of the results. If there in no statistical significance for the comparison of sub-groups, we cannot state that some results are higher/lower, as they are comparable (if there is no significance of differences, based on the statistical analysis the results are treated as the same). Taking it into account, the statements such as “decreased (but with no statistically significant differences)” should be in the whole manuscript corrected and changed into “comparable” or any word that is justified based on the conducted statistical analysis.

It seems that none of Authors is native English speaker, so some sentences are hard to follow and hard to understand (e.g. “Salt intake reduction is considered a priority adjustment in the world.” – it should be rather e.g. “Salt intake reduction is considered a public health priority”). The manuscript should be corrected by a professional English correcting agency.

Abstract:

Aim – should be properly formulated – “effects” – on what?

Authors should present here specific numeric results of the conducted study accompanied by a specific results of their statistical analysis (e.g. p-Values).

Authors should formulate any general conclusions – not being specific only for this studied population.

Introduction:

Authors should present international perspective, not being specific only for Japanese population/ Asian populations, but also the perspective for US population, and European populations.

Authors should properly use the references in this section – they should refer the presented information with the properly chosen literature. In the present version of the manuscript there are whole paragraphs with no references at all (e.g. lines 55-59).

Authors should not use this Section to present the results of their own previous studies, but rather to present the results of most important studies (!) from all countries.

Authors should properly formulate the aim of their study (e.g. “The aim of the study was…”) instead of presenting methods in this section (lines 64-67)

Materials and Methods:

Exclusion criteria should be defined precisely – were they formulated for all family members?

It is hard to believe that Authors did not have exclusion criteria associated with disturbances of sodium metabolism – were there any?

Any reference number (or date) of Ethical Committees agreement should be provided.

It is not explained by Authors why there was different number of participants in the studied groups – how did it happen if they were randomly allocated to groups?

It should be precisely described how was the sodium intake measured and in which dishes was it measured.

It seems that Authors did not verify the normality of distribution – they should verify it, and indicate which statistical test was used for verification.

For normally distributed data Authors should present mean and SD values, but for the other distributions – present median, min and max values

Authors should apply adequate statistical tests, that are based on the distribution.

Results:

It seems that Authors did not verify the normality of distribution – they should verify it, and indicate which statistical test was used for verification.

For normally distributed data Authors should present mean and SD values, but for the other distributions – present median, min and max values

Authors should apply adequate statistical tests, that are based on the distribution.

Instead of figures Authors should rather present tables.

Discussion:

Authors should correct the section accordingly, while discussing all the indicated problems.

References:

There is a problem of excessive self-citations, as Authors included over 18% of own references while some of them are not associated with the studied issue at all – such references should be removed. At the same time, Authors should include also references of other research teams, in order to present international perspective.

Round 2

Reviewer 3 Report

While the manuscript has improved in several areas and the readability is improved, there are still grammatical issues and a lack of clarity in some places.

Overall, I would avoid using Na as an abbreviation for sodium. It really focused on the sodium being an element and not a dietary nutrient. I recommend spelling out sodium throughout manuscript.

Abstract

Line 22 “for the changes as major outcomes”. It is not clear which specific changes the author is referring to

Line 25 should be non-significant, not “insignificant”. Nonsignificant is the appropriate statistical terminology to be used here

Line 26 should be twelve-months

Line 25-26 on urinary Na excretion returning to baseline is repeated on line 27. Report only once

Line 29. Should be restated that “reducing salt intake for short- and long-term among the people studied in this cohort”.

Intro

Up to line 28, the authors have used the term salt and now they introduce “sodium chloride”. Please be consistent

Line 60 I would not state “We think this is easy to implement”.

Line 60 should be “hypothesized”. Use past tense here. And why is the hypothesis here and now with the aim of the present study?

Line 67-68. What did these other studies find? Sentence seems unfinished

Figure 1 is blurry and still need to be improved.

Figure 2. While the numbers on the y axis are really big, the small font size of the axis titles and the legend for the groups seems out of place and strange. The y axis title is extremely hard to read.

Discussion

Line 338 should “in study”, not “on study”

A thought regarding the conclusion statement is that its more than just urinary Na excretion returning to baseline values. It is really that subjects returned to their old dietary habits.
